# Neural Activity Associated with Symptoms Change in Depressed Adolescents following Self-Processing Neurofeedback

**DOI:** 10.3390/brainsci12091128

**Published:** 2022-08-25

**Authors:** Natasha Ahrweiler, Carmen Santana-Gonzalez, Na Zhang, Grace Quandt, Nikki Ashtiani, Guanmin Liu, Maggie Engstrom, Erika Schultz, Ryan Liengswangwong, Jia Yuan Teoh, Katia Kozachok, Karina Quevedo

**Affiliations:** 1Department of Psychiatry, University of Minnesota, Minneapolis, MN 55455, USA; 2Department of Human Development and Family Sciences, University of Connecticut, Storrs, CT 06269, USA

**Keywords:** depression, neurofeedback, adolescents, self-processing network, neuroplasticity, self-processing, precuneus, middle temporal gyrus, cerebellum

## Abstract

Adolescent depression is prevalent, debilitating, and associated with chronic lifetime mental health disorders. Understanding the neurobiology of depression is critical to developing novel treatments. We tested a neurofeedback protocol targeting emotional regulation and self-processing circuitry and examined brain activity associated with reduced symptom severity, as measured through self-report questionnaires, four hours after neurofeedback. Depressed (*n* = 34) and healthy (*n* = 19) adolescents participated in (i) a brief neurofeedback task that involves simultaneously viewing their own happy face, recalling a positive autobiographical memory, and increasing amygdala-hippocampal activity; (ii) a self- vs. other- face recognition task with happy, neutral, and sad facial expressions before and after the neurofeedback. In depressed youth, reduced depression after neurofeedback was associated with increased self-referential and visual areas’ activity during neurofeedback, specifically, increased activity in the cuneus, precuneus and parietal lobe. Reduced depression was also associated with increased activation of emotional regulation and cross-modal areas during a self-recognition task. These areas included the cerebellum, middle temporal gyrus, superior temporal gyrus, and supramarginal gyrus. However, decreased rumination was linked to decreased precuneus, angular and temporal gyri activity during neurofeedback. These results tentatively suggest that neurofeedback may induce short-term neurobiological changes in the self-referential and emotional regulation networks associated with reduced symptom severity among depressed adolescents.

## 1. Introduction

Depression is highly prevalent in the adolescent population. From 2009 to 2017, rates of major depressive episodes increased by 63% in adolescents [1], and an estimated 17% of adolescents aged 12 to 17 were diagnosed with depression in 2020 [2]. The onset of depression most often occurs between ages 15 to 18 [3], perhaps due to the simultaneous increase in physiological (e.g., development of self-referential and emotional neural circuitries) and environmental (e.g., stressful life experiences) demands during the adolescent period [4]. Excessive self-referential processing accompanied by negative emotions is predictive of onset and chronicity of adolescent depression [5]. This critical period is related to increased symptom severity, including higher suicide risk [6]. Due to the high risks associated with this population, developing effective treatments is critical to improve adolescent health.

This study examines the relationship between brain activity, before, during, and after a neurofeedback task, and depression symptoms and rumination. The goal of this study was to (1) explore the neural activity linked to short-term symptom improvement following neurofeedback, and (2) explore brain areas of interest for future neurofeedback protocols.

### 1.1. Self-Referential Processing and Depression

Rumination entails repetitive thoughts, including dwelling on the causes and consequences of one’s behaviors and depressive symptoms [7]. Rumination accompanied by negative affect often manifests as unconstructive repetitive thoughts regarding one’s self-efficacy, leading to self-blame or self-criticism [5,8,9]. Studies of adults and adolescents show that those with prior depression diagnoses are more likely to exhibit rumination, and in turn, rumination is predictive of future symptoms [7,9,10]. Thus, rumination has important implications for emotion dysregulation and maladaptive self-processing [5].

Understanding the neurobiology of maladaptive rumination by focusing on the self-processing network, a subsection of the default mode network (DMN), may help develop effective treatments to reduce rumination in depression [10]. Moreover, researchers have pinpointed the interactions between the self-processing network and the salience and executive control networks [11], which are often also altered by rumination in depression [12,13,14]. Cortical midline structures subserve the self-processing network during introspective and self-relevant tasks [15]; the salience network identifies emotionally and cognitively salient information during attention-demanding tasks [14,16]; and the executive control network mediates goal-dependent attention and behavior (top-down processing) [17]. In healthy individuals, activation of the ‘task-positive circuits,’ such as the salience and executive control networks, occurs during attention-demanding tasks and is correlated with deactivation of the ‘task-negative networks,’ such as the self-processing network [18]. However, neuroimaging studies of depressed adults suggest that hyperactive cortical midline structures during attention-demanding tasks could mediate persistent rumination [10,19]. There is evidence of the relationship between hyperactive self-processing and depression in adolescence [10] and, as a period of significant corticolimbic development, more research is needed to understand the neurobiology of rumination in depressed adolescents [4,20]. We sought to investigate how cortical midline structural changes—including the medial dorsal and ventral prefrontal cortex, cingulate cortex, precuneus, and middle temporal gyrus—may associate with symptom improvement in depressed adolescents who underwent a neurofeedback procedure [10,21,22].

### 1.2. Emotional Regulation and Depression

The salience network is a large network encompassing cortical and limbic regions such as the amygdala, insula, and anterior cingulate cortex [23]. The salience network involves the amygdala-hippocampal complex (AMYHIPP) which is engaged by emotionally salient stimuli [16] and directing neural resources toward activation of the self-processing versus executive control networks [16,23].

In people with depression, salience attribution tends to perpetuate negative affect, and the amygdala is hyperactive during negative yet hypoactive during positive stimuli [24,25]. Adolescents with greater depression severity showed less activity in the AMYHIPP complex during positive self-processing (vs. neutral processing of others) tasks [12,26]. These data suggest that recruiting areas of the salience network during neurofeedback (e.g., AMYHIPP) could increase emotion regulation and mediate beneficial effects in the self-processing network.

### 1.3. Neurofeedback in Adult and Adolescent Populations

Neurofeedback has the potential to induce neural plasticity in targeted brain regions with beneficial effect. In healthy adults, neurofeedback using autobiographical memory recall has successfully increased hippocampus recruitment as well as connectivity between the hippocampus and amygdala in correlation with enhanced emotional regulation [27]. Young and colleagues administered neurofeedback to depressed adults and asked them to recall positive autobiographical memories while simultaneously increasing left amygdala engagement [13,28]. They measured increases in amygdala connectivity in association with reduced depression severity for an average of 9 days following the protocol, suggesting that brief neurofeedback may induce neuroplastic changes beyond the immediate time frame of the protocol [28,29,30].

Few studies on neurofeedback exist in adolescents. In adolescents with ADHD, Alegria et al. and Rubia et al. used real-time functional magnetic resonance imaging (fMRI) neurofeedback to effectively induce changes in neural activity which were correlated with symptom improvements [31,32]. One study on neurofeedback in depressed adolescents utilizing EEG yielded that both the real and sham neurofeedback treatments effectively reduced depression severity [33]. Compared to EEG, real-time fMRI has the advantage of providing feedback from loci in the brain with high spatial specificity, enabling investigation of functional relationships between neurological structures and cognitive functions [34].

The current study conducted a secondary data analysis and explored neural activity associated with short-term symptoms change during and after a real-time fMRI neurofeedback protocol in depressed adolescents. This study extends previous findings reported in Quevedo et al., 2019 and 2020. Specifically, Quevedo et al., 2019 investigated the overall network engagement during and before vs. after the neurofeedback task without exploring their association with symptoms change, whereas the current study focuses on exploring the network engagement that is associated with symptoms change. Quevedo et al. 2020 focused on amygdala functional connectivity and its links to symptom change during the task, while the current study focuses on neural activity, both during the neurofeedback task and during self-recognition tasks, and its association with symptoms change. As prior publications, this was a preclinical trial that aimed to garner preliminary data to create a foundation for a larger clinical trial. We hypothesized that, following a brief neurofeedback protocol that aimed to increase AMYHIPP activity during positively-valenced self-processing, depressed adolescents’ symptoms improvement would be associated with emotion regulation and self-processing networks engagement.

## 2. Materials and Methods

We recruited patients from the community and from outpatient and inpatient units at the University of Minnesota (U of M). Exclusion criteria were general magnetic resonance imaging exclusions, psychosis, and major medical and substance use disorders. The U of M Institutional Review Board (IRB) approved this study. During the first session, which were videotaped, we evaluated right-handed adolescents (*n* = 53, Table 1) using both the Kiddie Schedule for Affective Disorders and Schizophrenia Present and Lifetime Version, K-SADS-PL [35] and the continuous Children’s Depression Rating Scale, CDRS, [36] and IQ was sampled using the Wechsler Abbreviated Scale Intelligence [37]. A licensed clinical psychologist (KQ) diagnosed the presence or absence of depression during the first session by coding the videotapes of the first sessions, which included the formal structured clinical interviews via the K-SADS-PL. Depressed participants were stable on medication (Table 1). During the second session, the Mood and Feelings Questionnaire [38] and the Responses to Depression [39] were used to measure depression and rumination symptoms before and after scanning.

Before scanning, participants identified peak positive moments in 5–6 autobiographical memories and had pictures taken of their faces with happy, sad, and neutral expressions. Participants completed a short version of the Emotional Self-Other Morph Query task (ESOM-Q; Figure 1b) prior to the Emotional Self-Other Morph Neurofeedback task (ESOM-NF; Figure 1a) and again after neurofeedback.

The ESOM-Q task required participants to view images of their own face morphed with that of someone else’s. The images (*n* = 112) had a range from 60–100% of their own face features (High Self-Face) or 40–0% of their face features (Low Self-Face) and had either happy, sad, or neutral facial expressions yielding two (high self or low self) by three emotional expressions (happy, sad, neutral) blocks of images. For each face image, participants indicated whether it was a self or other face via a button press. Each 35-s block was followed by 18-s rest periods in the following order: happy self-face, happy other-face, neutral self-face, neutral other-face, sad self-face, sad other-face presented in three counterbalanced orders. The goal was to investigate the possible neurobiological changes in self vs. other face recognition before vs. after neurofeedback.

ESOM-NF is comprised of four 40-s blocks. First, participants saw their own happy face and were asked to recall their positive autobiographical memories verbally. Simultaneously, participants viewed a bar that displayed their AMYHIPP activity in real time provided by MURFI software [40]. AMYHIPP activity was green when activity was above baseline and red when activity was below baseline. While viewing the happy self-face and recalling positive autobiographical memories, they were asked to try and increase the activity of the AMYHIPP. This was followed by a control task, during which the participants saw an unfamiliar happy face and were asked to count backwards from 100. The participants were given three rests throughout the entirety of the task, for 30 s at onset, 12 s in the middle, and 12 s at the conclusion. After each neurofeedback and counting-backwards section, participants rated their mood (1 = bad to 4 = good). The goal of this task was to increase self-processing of positive emotions by pairing the autobiographical memory recall to happy self-face viewing.

Before scanning (time 1) after scanning (time 2), we administered Responses to Depression [39] and the Mood and Feelings Questionnaire, MFQ [38]. We selected rumination as maladaptive emotion regulation and self-referential strategies predictive of emergence and recurrence of depression [41,42,43], while the MFQ measured depression [38].

### 2.1. Online Analyses

MURFI software [40] generated amygdala and hippocampus activity values using subject specific anatomical masks of the bilateral amygdala and hippocampus. PsychoPy [44] was used to display the MURFI values as a visual thermometer bar during the ESOM-NF’s feedback condition. The bar representing amygdala and hippocampus values was updated as each new functional magnetic resonance imaging (fMRI) volume was acquired. Online subject head motion compensation was accomplished using the Siemens PACE/MoCo system [45]. Feedback automatically stopped if movement exceeded 4–3 mm repeatedly (which occurred in just one participant), but participants could re-initiate the Emotion Self-Other Morph Neurofeedback task. Region of interest [46] was localized anatomically during the multiband echo-planar imaging (EPI) series (target functional reference acquisition, see Appendix A) for each individual and mapped to individual’s T1 structural brain data. Data were collected using a 3.0 Tesla Siemens Prisma MRI scanner with the 32 Channel receive only head coil. Structural 3D axial MPRAGE images were acquired for each participant (TR/TE: 2100/3.65 ms; TI: 1100; Flip Angle 7°; Field of View: 256 × 256 mm; Slice-Thickness: 1 mm; Matrix: 256 × 256; 224 continuous slices), GRAPPA 2. Mean blood oxygen level dependent (BOLD) images were then acquired with a slice-accelerated gradient echo-planar imaging sequence during 6.08 min for the Pre- and Post- Emotion Self-Other Morph tasks and 6.02 min for the Emotion Self-Other Morph Neurofeedback task (2.4 mm^3^ voxels, covering 60 oblique axial slices; TR/TE = 1510/32.4 ms; FOV = 216 × 216 mm; matrix 90 × 90; Flip Angle 65°; multi-band acceleration factor 3).

### 2.2. Offline Analyses

We used SPM12 for preprocessing of functional magnetic resonance imaging (fMRI) data and all statistical analysis. Echo-planar imaging time series’ preprocessing included: (1) rigid body realignment for head motion correction, (2) slice timing correction, (3) rigid body co-registration of EPI with high resolution anatomical data, (4) spatial normalization to the Montreal Neurological Institute (MNI) anatomical space using unified segmentation, and (5) spatial smoothing (6 mm full width at half maximum). Head motion outliers in echo-planar imaging time series were identified and corrected using the Artifact Detection Tools with a scan-to-scan movement threshold of 0.5 mm and a scan-to-scan global signal change of 3 SD (www.nitrc.org/projects/artifact_detect/), accessed on 1 January 2020. For each subject, blood oxygen level dependent (BOLD) contrast signal variance was modeled with a set of regressors using a general linear model. The total signal variance was decomposed into a task component, with inter-trial intervals as implicit baselines. Each task regressor was constructed by generating condition duration vectors and then convolving them with a canonical hemodynamic response function, allowing parameter estimates proportional to task-related neural activity per second. The full model for each subject comprised: (1) the condition regressors, (2) regressors modeling movement-related signal modulation, (3) outlier time points, (4) the mean signal for the session, (5) a discrete cosine transform basis set that modeled the low frequency, presumably artifactual, signal modulations at frequencies lower than 0.008 Hz and (6) realignment and censoring regressors for nuisance physiological noise. Parameter estimates were calculated using a restricted maximum likelihood algorithm.

We conducted multiple regressions (*p_uncorr_* < 0.005) to examine brain activity during ESOM-NF feedback minus counting-backwards conditions as well as self-face versus other-face recognition during ESOM Post minus ESOM Pre tasks (ESOM-Q Post vs. ESOM-Q Pre). Specifically, the self vs. other face recognition contrasts for the ESOM task administered before neurofeedback were subtracted from the self vs. other recognition contrasts for the ESOM task administered after neurofeedback: Post- (self vs. other) minus Pre- (self vs. other) neurofeedback. These regressions included the following covariates: severity of depression at intake (measured via the CDRS), IQ, gender, medication load, rumination change (rumination at time 1 minus rumination at time 2) and depression change (depression at time 1 minus depression at time 2). Higher rumination or depression change (as per self-report questionnaires) values indicated symptom improvement, i.e., lower scores after the scanning procedures. However, here we report regression models exclusively in the depressed youth because only depressed youth (*n*= 34) showed significant reduction in depression and rumination symptoms after vs. before scanning procedures. Rumination change, *F*(1, 48) = 7.089, *p* < 0.05, and depression change, *F*(1, 48) = 17.389, *p* < 0.01 (Table 2). A combined voxel-height and cluster-extent threshold was calculated to control for Type 1 error using Monte Carlo simulations in AFNI (v. 18.2.06) (Cox, 1996). Using 3dClustSim, α = 0.01, *p* < 0.005 For ESOM-Pre vs. Post, smoothness estimates entered in 3dClustSim (9.68 9.89 9.36) were calculated by 3dFWHMx. Only clusters ≥ 93 voxels were significant for the ESOM task. For ESOM_NF, smoothness estimates were (10.38 8.96 10.24) and only clusters ≥ 97 were significant and thus reported.

To examine whether areas of activity overlapped between the two analyses, conjunction analyses were run between the maps from contrasts for ESOM_NF and for ESOM Pre vs. Post that were a cluster forming threshold of *p*_uncorr_ < 0.005.

## 3. Results

### 3.1. Brain Activity Associated with Symptom Change during Feedback vs. Counting Backwards

Lower rumination in depressed youth (*n* = 34) after vs. before scanning correlated with decreased engagement of the precuneus, angular gyrus, middle temporal gyrus, superior temporal gyrus, and inferior temporal gyrus during neurofeedback vs. counting-backwards (Table 2, Figure 2a).

In the depressed youth (*n* = 34), lower depression after vs. before scanning correlated with relative increased engagement of the cuneus, precuneus and parietal lobe (Brodmann Areas 7, 18, 19, and 31) during neurofeedback vs. counting-backwards conditions (Table 2, Figure 2b). Lower depression after vs. before scanning procedures was also associated with increased engagement of the lingual gyrus and inferior occipital lobe (Table 2, Figure 2b). Conjunction analyses showed that no area linked to symptom change overlapped with areas associated with depression severity at intake measured via the CDRS, suggestive of independent mechanisms for short-term symptoms decrease from the neural substrates of depression severity.

### 3.2. Brain Activity Associated with Symptom during Self-Processing

Lower depression was observed after vs. before scanning in depressed youth (*n* = 34) with increased engagement of bilateral cerebellum, middle temporal gyrus, frontal lobe, supramarginal gyrus, and superior temporal gyrus during self-processing (i.e., the self vs. other face recognition task) after vs. before neurofeedback (Table 2, Figure 3). Among depressed participants, rumination change did not significantly correlate with any change in brain activity during the self vs. other face recognition task. Brain areas activated for the post- vs. pre- neurofeedback analysis did not overlap with areas engaged during the neurofeedback vs. counting-backwards analysis.

## 4. Discussion

We sought to engage areas of emotion regulation during a self-processing task to induce changes in the self-processing and emotion regulation networks. Reduced depression and rumination levels were associated with changes in self-processing and emotion regulation networks as well as modulatory cross-modal areas in depressed participants. Specifically, increased engagement of multiple self-processing and regulatory areas during neurofeedback vs. count-backwards conditions and self-face recognition after versus before neurofeedback was linked to short term decrease in symptoms. This suggests that the balance in activity between the self-referential and executive control networks may be involved in depression neurophysiology and represent possible targets for depression and rumination symptoms improvement.

### 4.1. Symptoms Change and Neurofeedback

Short-term depression decrease in depressed youth was associated with increased cuneus, precuneus and parietal lobe activity during the neurofeedback vs. counting-backwards conditions. Research ties the precuneus to various functions; it is involved in visuo-spatial processes along with surrounding parietal areas, episodic memory retrieval, and self-processing [47,48,49]. With regards to self-processing, evidence suggests that the precuneus plays a strong role in retrieving self-relevant episodic memories and self-reflection [50,51]. Due to its involvement in episodic memory retrieval, participants who engaged more effectively with their autobiographical memories during neurofeedback might have recruited the precuneus more and subsequently experienced lower depression symptoms.

Research in fMRI neurofeedback has tied increased functional connectivity between the amygdala and neocortical regulatory areas with greater long-term symptom improvement [52]. These areas overlapped with the precuneus and posterior cingulate cortex, perhaps showing promise that the improved symptoms reported here might persist beyond the short time frame measured in this study, which was approximately 4–5 h between time 1 and time 2 of self-reported symptoms.

Reduced rumination in depressed youth correlated with decreased engagement of self-referential (precuneus, middle temporal gyrus, angular gyrus) processing in neurofeedback vs. counting-backwards tasks. The precuneus, middle temporal gyrus, and angular gyrus are key nodes of the self-processing network [48,53,54,55]. Imaging studies show volumetric differences in these areas associated with both treatment-responsive and treatment-resistant depression [56]. The association of rumination improvement with decreased engagement of several regions in the self-processing network is consistent with prior findings that depression is associated with hyperactive self-processing networks [9,10]. This self-processing network hyperactivity and increase in ruminative cognitions may limit the individual’s ability to process and integrate rewarding, self-relevant information [57,58]. Therefore, decreasing activity in these areas through neurofeedback may mediate improvement of rumination symptoms.

Additionally, there was increased engagement of visual areas during neurofeedback associated with either depression (cuneus, inferior occipital lobe, BA 7, BA 18, BA 19) or rumination (middle occipital gyrus) improvement. The role of the cuneus, middle temporal gyrus, angular gyrus, and middle occipital gyrus in attention modulation and reward anticipation of visual stimuli [47], as well as the lingual gyrus, inferior occipital lobe, and posterior cingulate cortex (BA 31) in facial recognition and processing [59,60,61,62] cortical areas (involved in visual attention modulation) and their interaction with limbic circuits during self-processing might be a mechanism for symptom improvement.

### 4.2. Post- vs. Pre- Neurofeedback Self Processing and Symptoms Improvement

During the post- vs. pre- neurofeedback self-processing task, specifically self vs. other face recognition after versus before the neurofeedback task, youth that reported reduced depression symptoms showed an increase in self-processing networks’ activity, including the middle temporal gyrus after vs. before neurofeedback during the self-recognition task. Along with being a key area for self-processing, the middle temporal gyrus is a cross-modal hub with overlap between conceptual and semantic processing [63] Davey et al. posited that the middle temporal gyrus participates in both intrinsic, self-referential processing as well as attention-demanding tasks, perhaps representing a nexus between the self-referential and executive demand networks which are otherwise anti-correlated [64]. Our results indicate that increasing recruitment of this area during self-processing (operationalized as self vs. other face recognition) is associated with decreased self-reported depression symptoms. Given the decreased middle temporal gyrus activity among depressed individuals [56], perhaps increased coordination between the executive control and self-processing networks is associated with decreased depression symptoms.

Increased engagement of the superior temporal gyrus was also associated with decreased depression. This, along with activation of the supramarginal gyrus, implicates anatomical portions of the inferior parietal lobule/temporal parietal junction, which support self-referential, executive control, and emotional salience [65,66]. The temporal parietal junction mediates self vs. other emotional processing and attention modulation between introspection and the environment [65]. Increased engagement at this area may indicate an enhanced regulatory mechanism between introspective vs. environmental attention.

The analysis also indicated that increased bilateral cerebellum recruitment is associated with lower depression symptoms. The cerebellum supports motor processes but is also involved in higher order cognitive functions such as affective processing, social cognition, and self-reflection [67,68,69]. It is involved in self and social processing, such as facial emotion recognition, empathy, and abstract mentalization: conceptualizing one’s future self and autobiographical past [46,69]. Although the cerebellum’s behavioral correlates are a growing area of research, reviewers hypothesize that it may not have a unique function itself, but is instead a contributor of adaptive feedback to various other structures—evidenced by increased cerebellum activity associated with increased difficulty of various motor, cognitive, and emotional tasks [69]. The association between cerebellum activity during self-processing and improved depression scores might indicate several functions, such as enhanced emotion regulation, mentalization, or self-reflection.

### 4.3. Limitations and Future Directions

This evidence from this study provides important implications for an increasingly interesting area of depression research. Understanding the mechanisms of the neural plasticity following neurofeedback will give a foundation for future clinical research, and this research provided an innovative approach to this topic. However, there are also several limitations to our study. This study recruited a relatively small sample size of depressed youth. Future studies should attempt to recruit a larger cohort. Additionally, the improvement of symptoms could be due to factors other than the neurofeedback, such as recall of the positive memory alone or the mere act of being in an MRI machine. Because there is not a placebo group, changes in brain activity reported here could be a regression toward the mean. We also suspect that there was no change seen in controls due to a floor effect in our questionnaire measurements. Future studies should have a sham version for AMYHIPP activity that is non-contingent on the task at hand, or a placebo neurofeedback target unrelated to the mental activity at hand, enabling researchers to draw reasonable causality between the tasks and the changes in brain activity.

Future research should also focus on the longevity of symptom improvement and neuroplasticity in association with neurofeedback specifically in depressed adolescents. Our study measured the changes in symptoms within hours of the tasks, indicating a potential for short-term change but not necessarily long-term improvements. As mentioned, research has indicated that increased connectivity between the amygdala and areas of interest here (precuneus and posterior cingulate cortex) predicts long-term symptom improvement [52]. Furthermore, a study of repetitive transcranial magnetic stimulation for treatment-resistant depression found that increased functional connectivity between hubs of the self-processing network conferred greater clinical improvement at a three-month time-point [70]. In the case of neurofeedback, there is evidence that even a single session may make an individual more cognizant of emotional regulation strategies and their efficacy, and thus they will be employed more for symptomatic improvement beyond the immediate timeframe [71]. Nevertheless, more research is needed regarding the number and length of neurofeedback sessions required to instill long-term change.

## 5. Conclusions

This study identifies brain areas of interest for short-term symptom improvement in a novel fMRI neurofeedback study of depressed adolescents. These results implicate areas important to self-referential functions as well as areas that are modulators in activation of the self-processing network vs. executive control network—circuitry that is anti-correlated in healthy controls. The results suggest the precuneus, middle temporal gyrus, and angular gyrus as possible areas of the self-processing network that are hyperactive in depressed youth. Both analyses showed increases in activity at cross-modal hubs between networks, including the inferior parietal lobule/temporal poles and the middle temporal gyrus, in association with improved symptoms. The cerebellum also showed increased activity after the neurofeedback protocol, which may be linked to changes in emotional regulation during self-referential processing. or perhaps beneficial modulation to the self-referential circuitry. Overall, the results suggest that engagement of emotion regulation at the amygdala-hippocampal complex could elicit neural plasticity in areas of self and emotional processing in association with improvement of depression and rumination symptoms. This both corroborates the involvement of self-referential processing in adolescent depression and indicates potential therapeutic targets for prospective studies. This study provides preliminary data regarding the potential for neurofeedback to remedy symptoms in depressed youth.

## Figures and Tables

**Figure 1 brainsci-12-01128-f001:**
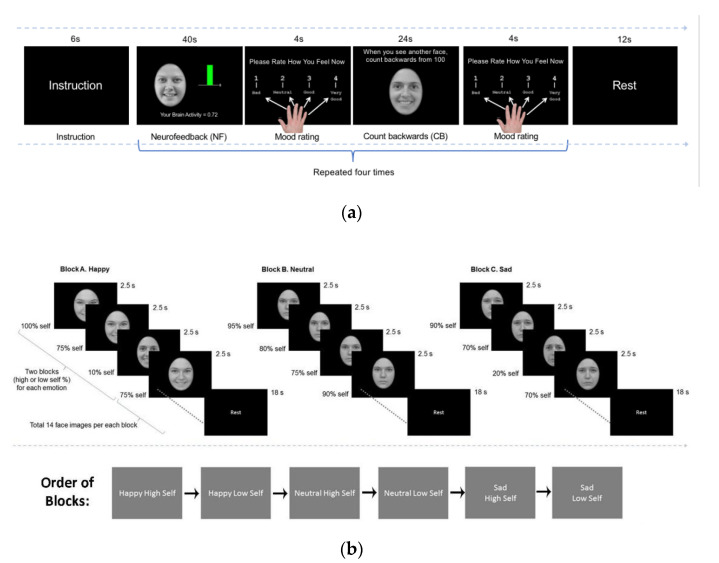
Task Structures: (**a**) Emotional Self-Other Morph Neurofeedback Task (ESOM-NF) entailed participants recalling positive autobiographical memory to the cue of their own smiling face for 40 s and attempting to increase the green bar (the amygdala-hippocampal complex), viewing another’s face and counting backward for 24 s, rating their mood for 4 s after feedback or count backward blocks, and resting for 12 s. (**b**) Emotional Self-Other Morph-Query (ESOM-Q) was administered before (ESOM-Pre) and after (ESOM-Post) the neurofeedback task (ESOM_NF). Participants recognized faces as either their own or as different face via button press.

**Figure 2 brainsci-12-01128-f002:**
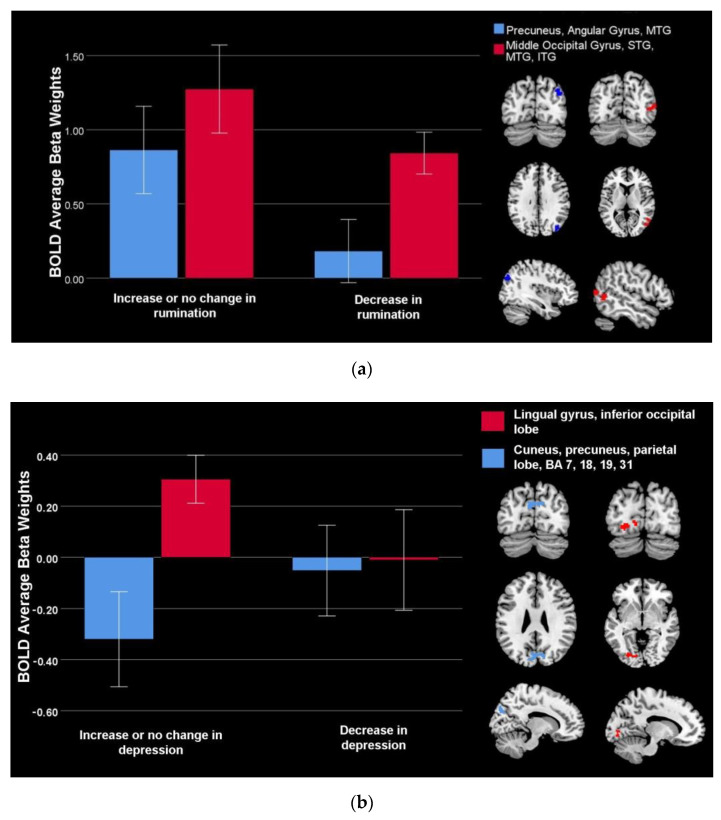
Brain activity associated with symptom change during neurofeedback (feedback vs. count—backwards) in depressed youth, *n* = 34: (**a**) Changes in brain activity during neurofeedback vs. counting backward associated with decreased rumination; (**b**) Changes in brain activity during neurofeedback vs. count—backwards associated with decreased depression.

**Figure 3 brainsci-12-01128-f003:**
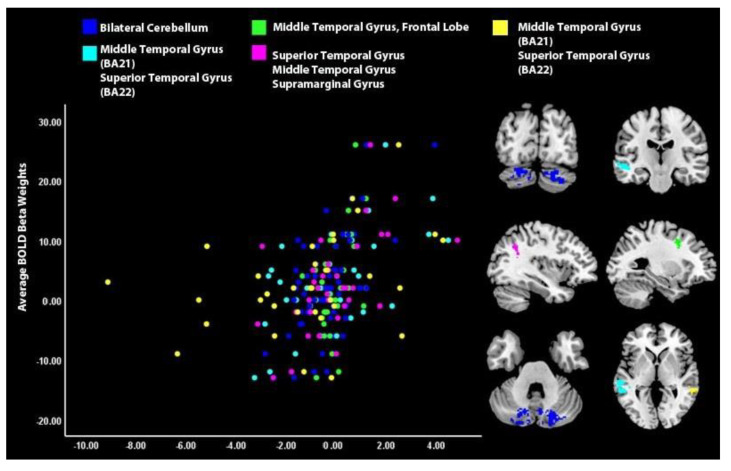
Brain activity during self-processing associated with symptom change in depressed youth, *n* = 34: Changes in activity during ESOM-Q Post vs. ESOM-Q Pre were associated with change in depression score.

**Table 1 brainsci-12-01128-t001:** Demographics and clinical presentation by diagnostic group.

	Healthy Controls	Depressed
	*n* = 19	*n* = 34
**Suicide attempters**	*n* = 0 a	*n* = 15 b
**Age at Intake: M (SD)**	16.26 (1.19)	16.08 (1.27)
**Age at Scanning: M (SD)**	16.35 (1.23)	16.11 (1.25)
**IQ:M (SD)**	115.32 (9.12) a	108.35 (10.84) b
**Sex**		
Male	7 (36.84%)	10 (29.41%)
Female	12 (63.16%)	24 (70.59%)
**Puberty: M (SD)**	4.53 (0.65)	4.53 (0.68)
**Ethnicity** White	14 (73.68%)	27 (79.41%)
African American/Black	0	2 (5.88%)
Native American	0	2 (5.88%)
Asian	3 (15.79%)	0
Other Ethnicity	2 (10.53%)	3 (8.82%)
**Family Structure** Married	15 (78.95%)	22 (64.71%)
Living with partner	1 (5.26%)	3 (8.82%)
Separated-Divorced	3 (15.79%)	5 (14.71%)
Single	0	4 (11.76%)
Income		
≥ 35K	0	6 (17.65%)
35–75 K	7 (36.84%)	9 (26.47%)
+ > 75 K	12 (63.16%)	19 (55.88%)
**Depression before neurofeedback: M (SD)**	3.76 (3.95) a	29.73 (13.79) b
**Depression after neurofeedback: M (SD)**	2.26 (2.46) a	20.35 (16.62) b *
**Rumination before neurofeedback: M (SD)**	29.31 (6.97) a	50.71 (11.68) b
**Rumination after neurofeedback: M (SD)**	27.89 (7.25) a	44.06 (14.76) b *
**Depression Severity (CDRS): M (SD)**	19.21 (3.56) a	49.85 (16.14) b
**Depression Diagnosis (K-SADS-PL)**		
**Major Depressive Disorder (MDD)**	0	14
**MDD with Psychotic Features**	0	1
**Dysthymia**	0	4
**Melancholic Depression**	0	1
**Depressive Disorder-NOS**	0	15
**Eating Disorders (K-SADS-PL)**	0	2
**Anxiety Disorders (K-SADS-PL)**	0	22
**PTSD (K-SADS-PL)**	0	6
**Disruptive Behavior Disorders (K-SADS-PL)**	0	6
**Substance Use Presence (K-SADS-PL)**	0	2
**Medication**		
Antidepressants	0	26
Antipsychotics	0	2
Mood stabilizers	0	0
Anxiolytics	0	10

Note: a and b denote significant differences between the compared means within groups or between groups across study time points. * Denotes *p* < 0.05. M = Mean and SD = Standard Deviation.

**Table 2 brainsci-12-01128-t002:** Whole brain results in depressed youth only.

	Direction of Prediction	Voxels	Hemisphere	MNI Coordinates			T
				X	Y	Z	
**Areas linked to depression change during neurofeedback vs. counting-backwards (ESOM-NF).**							
Lingual gyrus, inferior occipital lobe	Positive	112	Left	−28	−80	−6	3.58
Cuneus, precuneus, parietal lobe, BA 7, 18, 19, 31	Negative	276	Bilateral	−4	−82	26	4.06
**Areas linked to rumination change during neurofeedback vs. counting backwards (ESOM-NF).**							
Middle occipital gyrus, superior temporal gyrus, inferior temporal gyrus, middle temporal gyrus, BA 19, 37, 39	Positive	152	Right	52	−58	6	4.40
Precuneus, angular gyrus, middle temporal gyrus, BA 19 and 39	Positive	98	Right	38	−82	34	4.81
**Areas linked to depression change during self-recognition post- vs. pre- neurofeedback (ESOM-Post minus ESOM-Pre).**							
Cerebellum	Positive	429	Right	24	−80	−26	5.07
Middle temporal gyrus (BA 21), superior temporal gyrus (BA 22)	Positive	547	Left	−64	−26	−2	4.92
Frontal lobe, middle frontal gyrus	Positive	121	Left	−18	−10	42	4.50
Cerebellum	Positive	283	Left	−18	−78	−42	4.50
Supramarginal gyrus, middle temporal gyrus, superior temporal gyrus	Positive	127	Left	−38	−54	32	3.94
Middle temporal gyrus (BA 21), superior temporal gyrus (BA 22)	Positive	111	Right	68	−40	0	3.79

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
