# Peer review of "Neural Activity Associated with Symptoms Change in Depressed Adolescents following Self-Processing Neurofeedback"

_brainsci, 2022, doi:10.3390/brainsci12091128_

Round 1

Reviewer 1 Report

The study is of interest with potentially important implications, there are some concerns regarding the study design.

The sample size for the control group is very small (N=19). It would benefit the study to be presented as a pilot study (part of a larger ongoing study).

In materials and methods section authors present that “A licensed clinical psychologist (KQ) diagnosed the presence or absence of depression during the first session”. It would be helpful to use the K-SADS in the clinical interview.

Authors must add in limitation section the small sample of control group.

Author Response

Reviewer 1

Is the research design appropriate? Can be improved: The study is of interest with potentially important implications, there are some concerns regarding the study design.

The sample size for the control group is very small (N=19). It would benefit the study to be presented as a pilot study (part of a larger ongoing study).

Authors: The healthy control participants did not show a significant change in symptoms and because of that they were excluded from this scientific report. However prior publications by Dr. Quevedo did include the 19 healthy control participants (Quevedo et al., 2019; Quevedo et al., 2020).

In materials and methods section authors present that “A licensed clinical psychologist (KQ) diagnosed the presence or absence of depression during the first session”. It would be helpful to use the K-SADS in the clinical interview.

Authors: We did use the K-SADs in the clinical interview, it was video recorded and coded and used for diagnosis by the last author KQ. Please see our methods sections where the K-SADS-PL is referred to.

Authors must add in limitation section the small sample of control group.

Authors: We have indeed added the small sample size to the liminations.

References

Quevedo, K., Liu, G., Teoh, J. Y., Ghosh, S., Zeffiro, T., Ahrweiler, N., Zhang, N., Wedan, R., Oh, S., Guercio, G., & Paret, C. (2019). Neurofeedback and neuroplasticity of visual self-processing in depressed and healthy adolescents: A preliminary study. Developmental Cognitive Neuroscience, 40, 100707-100707. https://doi.org/https://doi.org/10.1016/j.dcn.2019.100707

Quevedo, K., Teah, J. Y., Engstrom, M., Wedan, R., Santana-Gonzalez, C., Zewde, B., Porter, D., & Cohen Kadosh, K. (2020). Amygdala Circuitry During Neurofeedback Training and Symptoms’ Change in Adolescents With Varying Depression. Frontiers in Behavioral Neuroscience. https://doi.org/https://doi.org/10.3389/fnbeh.2020.00110

Reviewer 2 Report

In this manuscript, the authors investigated the association between neural activity and symptom change in depressed adolescents following self-processing neurofeedback. There is increasing interest in the antidepressant effect of neurofeedback and the underlying neural mechanism and the current manuscript may contribute to this field of literature. The manuscript, however, needs major revision in terms of several important issues.

1), abstract, lines 18-20: the meaning of "reduced depression" is unclear because the authors did not explain the design of the study nor did they specify if reduced depression means lower depression or decreasing depression after neurofeedback. lines 19-20, grammar issue.

2), methods: why do the authors think the current sample size is enough for the current analysis? the authors stated that "a licensed clinical psychologist diagnosed the presence or absence of depression", so what is the diagnosis criteria used? section 2.1 and 2.2, please explain what is online versus offline analysis.

3), results: figures 1 and 2 are hard to read, therefore it is impossible to judge if the design of the tasks is appropriate. 

Author Response

Reviewer 2

Is the research design appropriate? Can be improved

Are the methods adequately described? Must be improved

Are the results clearly presented? Must be improved

Are the conclusions supported by the results? Can be improved

In this manuscript, the authors investigated the association between neural activity and symptom change in depressed adolescents following self-processing neurofeedback. There is increasing interest in the antidepressant effect of neurofeedback and the underlying neural mechanism and the current manuscript may contribute to this field of literature. The manuscript, however, needs major revision in terms of several important issues.

1), abstract, lines 18-20: the meaning of "reduced depression" is unclear because the authors did not explain the design of the study nor did they specify if reduced depression means lower depression or decreasing depression after neurofeedback. lines 19-20, grammar issue.

Authors: We have edited the abstract and added the requested information.

2), methods: why do the authors think the current sample size is enough for the current analysis? the authors stated that "a licensed clinical psychologist diagnosed the presence or absence of depression", so what is the diagnosis criteria used?

Authors: A licensed psychologist reviewed videotapes of the K-SADS to establish the final diagnosis of the participants including absence and presence of depression. This has been made more clear in the methods sections. Thank you for pointing out that this was not clear in the initial version.

section 2.1 and 2.2, please explain what is online versus offline analysis.

Authors: Online analysis pertain to online acquisition of activity in the hypocampus and amygdala that ocurr in real time and that are enabled by MURFI software as noted in the paper. Off-line analyses ocurr post-hoc after the neurofeedback session has finlized, it pertains to analyses that take place with neuroimaging data gathered once the session is over. Offline analyses took place with spm12 as noted in the methods section 2.2.

3), results: figures 1 and 2 are hard to read, therefore it is impossible to judge if the design of the tasks is appropriate.

Authors: We have reformatted the figures in the hope that the results will now show better.

Reviewer 3 Report

Thank you for the opportunity to review your work. The work presents the topic extensively. The introduction of the work is comprehensive, the purpose of the study and the hypotheses are included at the end. The methodological part is described flawlessly, the reader can understand the authors' research intention. The results are presented in the form of table descriptions and figures, I have no objections to the presentation of data and statistical inference. The discussion of the work stands at a high level. I would ask the authors to describe in the "limitations" section the strengths of the study - you can point to factors that demonstrate the pioneering nature of the project and fill the research gap.

Regards

Author Response

Reviewer 3

Are the conclusions supported by the results? Can be improved

Thank you for the opportunity to review your work. The work presents the topic extensively. The introduction of the work is comprehensive, the purpose of the study and the hypotheses are included at the end. The methodological part is described flawlessly, the reader can understand the authors' research intention. The results are presented in the form of table descriptions and figures, I have no objections to the presentation of data and statistical inference. The discussion of the work stands at a high level. I would ask the authors to describe in the "limitations" section the strengths of the study - you can point to factors that demonstrate the pioneering nature of the project and fill the research gap.

Authors: Thank you for your feedback  we have added some information to the conclusions to highlight the innomative nature of the research.

Round 2

Reviewer 2 Report

Thank the authors for addressing my concerns.